# Graphite phase carbon nitride based membrane for selective permeation

Yang Wang[1], Niannian Wu[1], Yan Wang[1], Huan Ma[1], Junxiang Zhang[1], Lili Xu[1], Mohamed K. Albolkany[1] & Bo Liu [1]

Precise control of interlayer spacing and functionality is crucial in two-dimensional material based membrane separation technology. Here we show anion intercalation in protonated graphite phase carbon nitride (GCN) that tunes the interlayer spacing and functions of GCN-based membranes for selective permeation in aqueous/organic solutions. Sulfate anion intercalation leads to a crystalline and amphipathic membrane with an accessible interlayer spacing at ~10.8 Å, which allows high solvent permeability and sieves out the solutes with sizes larger than the spacing. We further extend the concept and illustrate the example of GCN-based chiral membrane via incorporating (1R)-(-)-10-camphorsulfonic anion into protonated GCN layers. The membrane exhibits a molecular weight cutoff around 150 among various enantiomers and highly enantioselective permeation towards limonene racemate with an enantiomeric excess value of 89%. This work paves a feasible way to achieve water purification and chiral separation technologies using decorated laminated membranes.

---

[1] Hefei National Laboratory for Physical Sciences at the Microscale, Fujian Institute of Innovation of Chinese Academy of Sciences, School of Chemistry and Materials Science, University of Science and Technology of China, Hefei, Anhui 230026, China. Correspondence and requests for materials should be addressed to B.L. (email: liuchem@ustc.edu.cn)

Porosity of membranes comprised of two-dimensional (2D) materials originates from the intrinsic pores of layers and/ or the spacing in-between layers ($d$ value)[1–5]. The dense stacking of 2D layers typically renders a $d$ value around 3–4 Å, which restricts most molecules to access. Therefore, tuning of $d$ value is critical to advance the sieving performance[6–8]. Introducing water layers in between 2D GO laminates can change the spacing, however, the structure is susceptible to the atmospheric humidity[9]. Coating with hydrophobic layer helps to lock and stabilize the water layers in GO laminates and makes the $d$ value reliable for ion sieving[10]. By virtue of the strong cation-π interactions, the interlayer spacing of GO membrane can be accurately regulated for cation sieving[11]. The adjustable $d$ value can be achieved via partial exfoliation of graphite phase carbon nitride (GCN) material, but it is sensitive to the preparation procedures[12]. Recent works also demonstrated GCN-based membrane for micro-, ultra- and nano-filtration, as well as forward osmosis and pervaporation via compositing with various functional materials[13–18]. In spite of the impressive progress in membrane comprised of 2D layers, it remains challenging to assemble multifunctional membranes with tunable and stable $d$ value for task-specified applications, such as bio-molecules sieving and separation in non-aqueous environment[7].

In comparison to GO with uneven and random distribution of oxygenous groups, the lone-pair electron on proportioned $sp^2$ nitrogen makes GCN apt to be protonated and soluble in strong acidic medium[19]. As a consequence, corresponding anions and protonated GCN sheets are gathered together via electrostatic interaction. The GCN-acid composite with acid radical intercalated in the protonated GCN laminates can be precipitated when adding anti-solvent into GCN-acid solution. Note that the chemically inert 2D hexagonal boron nitride (h-BN) can't be protonated even using concentrated $H_2SO_4$[20]. The soluble, chemically stable and particularly protonation-feasible attributes entitle GCN more opportunities for preparing sieving membranes with unique properties. On one hand, anion mediated in the protonated GCN sheets renders robust sandwich structure owing to the strong electrostatic interaction. On the other hand, using Brønsted acids with diverse sizes and functionalities, it is facile to control the interlayer spacing at molecular precision and introduce desired functionality into the interlayer space, i.e., chirality.

Here we demonstrate the anion intercalation strategy for GCN functionalization with desired performances (Fig. 1a). The sulfate anion that intercalates into protonated GCN layers (denoted as GCN-SA, SA = sulfuric acid) enables stable structure of the composite system, accompanied with accurately controlled $d$ value that is enlarged by 10.8 Å. The amphipathic GCN-SA membrane contributes to high permeability of solvents with varied polarities. The species with different sizes can permeate through the membrane, while the solutes with hydrated radii larger than 5.4 Å are completely blocked, thus realizing precise sieving at sub-nanometer scale (Fig. 1b). Furthermore, the intercalated (1R)-(−)-10-camphorsulfonic acid (CSA) anion simultaneously tailors the $d$ value and creates chiral sites in-between protonated GCN layers (Fig. 1c). The assembled GCN-CSA membrane that is applicable in both aqueous and organic solutions, shows high enantioselective permeation efficiency, upon which the enantiomeric excess (ee) value of limonene racemate can reach high up to 89%.

## Results

**Structure and amphipathic property of GCN-SA membrane.** The detailed preparation procedures of GCN-SA composite are given in Methods section, the microstructure, chemical structure and composition are also analyzed and presented (Supplementary Note 1, Supplementary Figs. 1–6, and Supplementary Table 1). GCN-SA membranes consisting of sulfate ions in the protonated GCN layers were deposited onto varied substrates by vacuum filtration method ("Methods" section). The SEM images show a continuous and smooth membrane without identifiable pinholes and cracks on mixed cellulose esters (MCE, pore size: 200 nm, porosity: ~50%) substrate (Fig. 2a). The thickness of the membrane is tunable depending on the amount of GCN-SA. A typical membrane with thickness of ~700 nm is displayed in Fig. 2b and c. In comparison with GCN membrane, the (002) diffraction peak at 27.6° disappears in GCN-SA membrane; whereas new diffraction peaks at 6.28° (001n) and 12.42° (002n) appear, indicating a new phase with an interlayer spacing of 14.06 Å was produced (Fig. 2d). This is consistent with the XRD analysis of GCN-SA powder sample (Supplementary Fig. 3). Taking $d$ value of 3.26 Å of GCN into account[21], the intercalation of sulfate ion increases the $d$ value by ~10.8 Å, which is applicable for molecule/ion accessing. There is no swelling effect observed

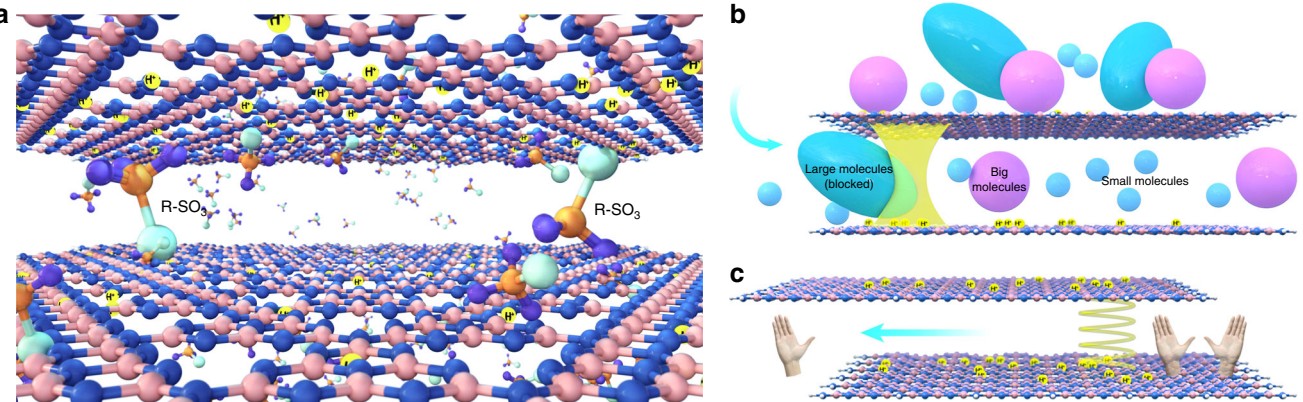

**Fig. 1** Schematic illustration of GCN functionalization for selective permeation. **a** Precise interlayer distance control of protonated GCN with desired functionalities via anion intercalation strategy. R-SO$_3$ represents the anion of Brønsted acid or chiral organic acid, sulfate or (1R)-(-)-10-camphorsulfonate anion is utilized in this work. Orange, blue and azure balls in R-SO$_3$ denote S atom, O atom and remaining group in acid, respectively. Pink, dark blue and light yellow balls in GCN framework denote N atom, C atom and proton, respectively. **b** Sieving effect of solutes with different sizes over GCN-SA membrane. Light blue ellipse, pink and light blue balls denote large, big and small molecules, respectively. The yellow spring denotes the interlayer space in-between protonated GCN layers. **c** Enantioselective permeation effect of different enantiomers over GCN-CSA membrane. The yellow spiral denotes the chiral sites created in-between protonated GCN layers, the left and right hands denote a pair of enantiomer for enantioselective permeation. The arrows in **b** and **c** denote the permeation direction of various solutes

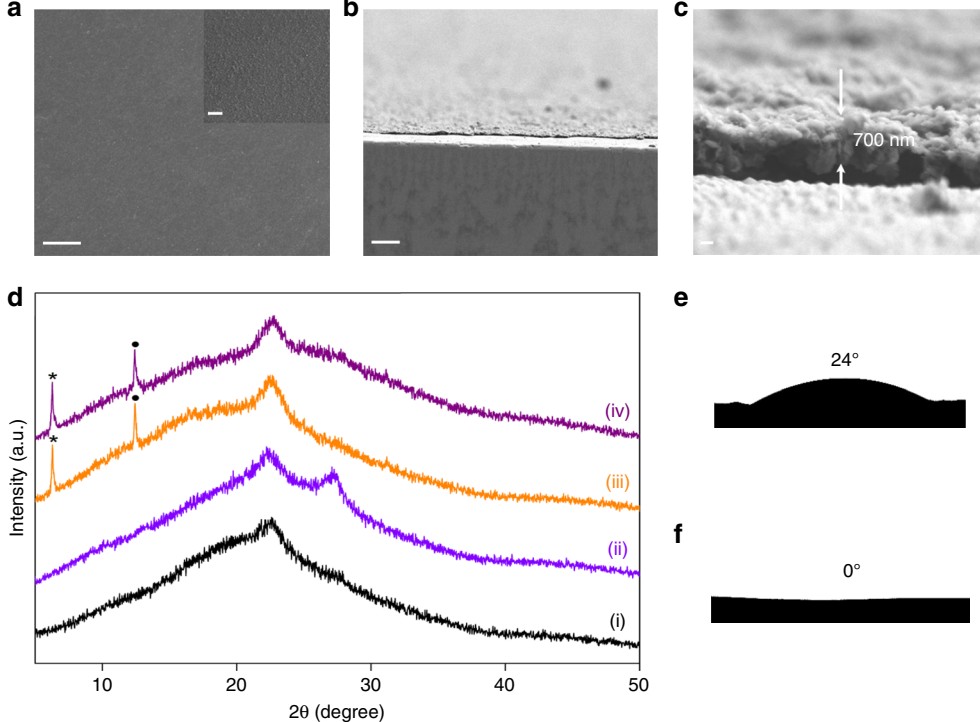

**Fig. 2** Characterizations of GCN-SA membrane. **a–c** Top-view (**a**) and cross-sectional (**b**, **c**) SEM images of GCN-SA membrane. **d** X-ray diffraction patterns of (i) blank MCE substrate; (ii) dried GCN membrane on MCE; (iii) dried GCN-SA membrane on MCE; (iv) dried GCN-SA membrane on MCE after immersing in water for 3 days. Peaks marked as black stars and balls denote the diffraction peaks (001n) and (002n), respectively, n denotes the new phase. **e**, **f** Digital photos of water drop (**e**) and cyclohexane drop (**f**) on CN-SA membrane. Scale bars for **a**, **b**: 10 μm, inset of **a**: 1 μm and **c**: 200 nm

when re-immersing the dried membrane in water, benefiting from the strong electrostatic interaction between sulfate ions and the protonated GCN sheets as discussed above (Fig. 2d). This is a sharp contrast to the susceptive $d$ value in water-mediated GO membrane[9] and partially exfoliated GCN membrane[12].

The contact angle of the GCN-SA membrane to water (24°, Fig. 2e) is evidently lower than that of GCN membrane (47°, Supplementary Fig. 7), suggesting its hydrophilic nature owing to protonation of GCN in GCN-SA membrane and assuring the excellent dispersion of GCN-SA in water. Further test using nonpolar solvent of cyclohexane gives rise to a near-zero contact angle (Fig. 2f). These results reveal the amphipathic property of GCN-SA membrane, which could greatly extend the application of GCN-SA membrane not only in aqueous but also organic solvent system.

**Solvent permeability of GCN-SA membrane**. As shown in Fig. 3a, the water permeability of GCN-SA membrane on MCE decreases with the increasing membrane thickness (see "Methods" section, Supplementary Fig. 8, and Supplementary Table 2). The GCN-SA membrane with a thickness of 700 nm readily reaches a water permeability of 104 L m$^{-2}$ h$^{-1}$ bar$^{-1}$, ~9 folds of that for GCN membrane (thickness: 500 nm, 11.1 L m$^{-2}$ h$^{-1}$ bar$^{-1}$). The water permeability on GCN membrane is ascribed to the passage generated by the existence of disordered GCN nanosheets according to the report in literature[22]. Under the identical test conditions, the blank MCE substrate shows negligible barrier with a very high water permeability of 5218 L m$^{-2}$ h$^{-1}$ bar$^{-1}$. We further managed to determine the permeability of a number of solvents with different polarities over GCN-SA membrane supported on PTFE substrate (Polytetra-fluorothylene, pore size: 200 nm, porosity: ~50%) (Fig. 3b), as MCE substrate is not stable in organic solvents. Organic solvent

permeability tests over blank PTFE substrate indicate that various solvents are free to pass through. The GCN-SA membrane on PTFE shows a water permeability of 111 L m$^{-2}$ h$^{-1}$ bar$^{-1}$, comparable with that of GCN-SA membrane on MCE. The membrane is permeable over solvents with a wide range of polarity. The most hydrophilic liquid of water shows highest permeability and the most hydrophobic liquid of cyclohexane shows the lowest permeability. Among these, other organic solvents, such as methanol and dioxane, also show favorable solvent permeability despite of the more hydrophobic property in comparison with that of water. The combined findings support the amphipathic property of GCN-SA. Also, we experimentally find that GCN-SA is capable of highly dispersing in organic solvents, such as ethanol, methanol, IPA, etc., distinguishing it from pristine GCN which can only poorly disperse in most common solvents upon long-time sonication[23]. We ascribe this to the structural attributes of GCN-SA, in which the conjugated basal plane is apparently hydrophobic, while the oxygen-containing groups induced by SA functionalization can endow GCN-SA with hydrophilcity[24]. The solvent permeability decreases with decreasing polarity, which is explained as the friction increasing with decreasing polarity of solvents in amphipathic interlayer space of GCN-SA membrane[25]. Moreover, the negligible cyclohexane permeability proves that the membrane is continuous, crack- and pinhole-free as well as non-mechanical leaking in the setup.

**Sieving performance of GCN-SA membrane**. Permeation tests were conducted using an isobaric setup and various technics were employed to monitor the concentration changes in both feed and permeate compartments depending on the properties of solutes (see "Methods" section, Supplementary Notes 2 and 3, Supplementary Figs. 8–12, and Supplementary Tables 3–5). GCN-SA membrane with thickness of ~700 nm was used for permeation

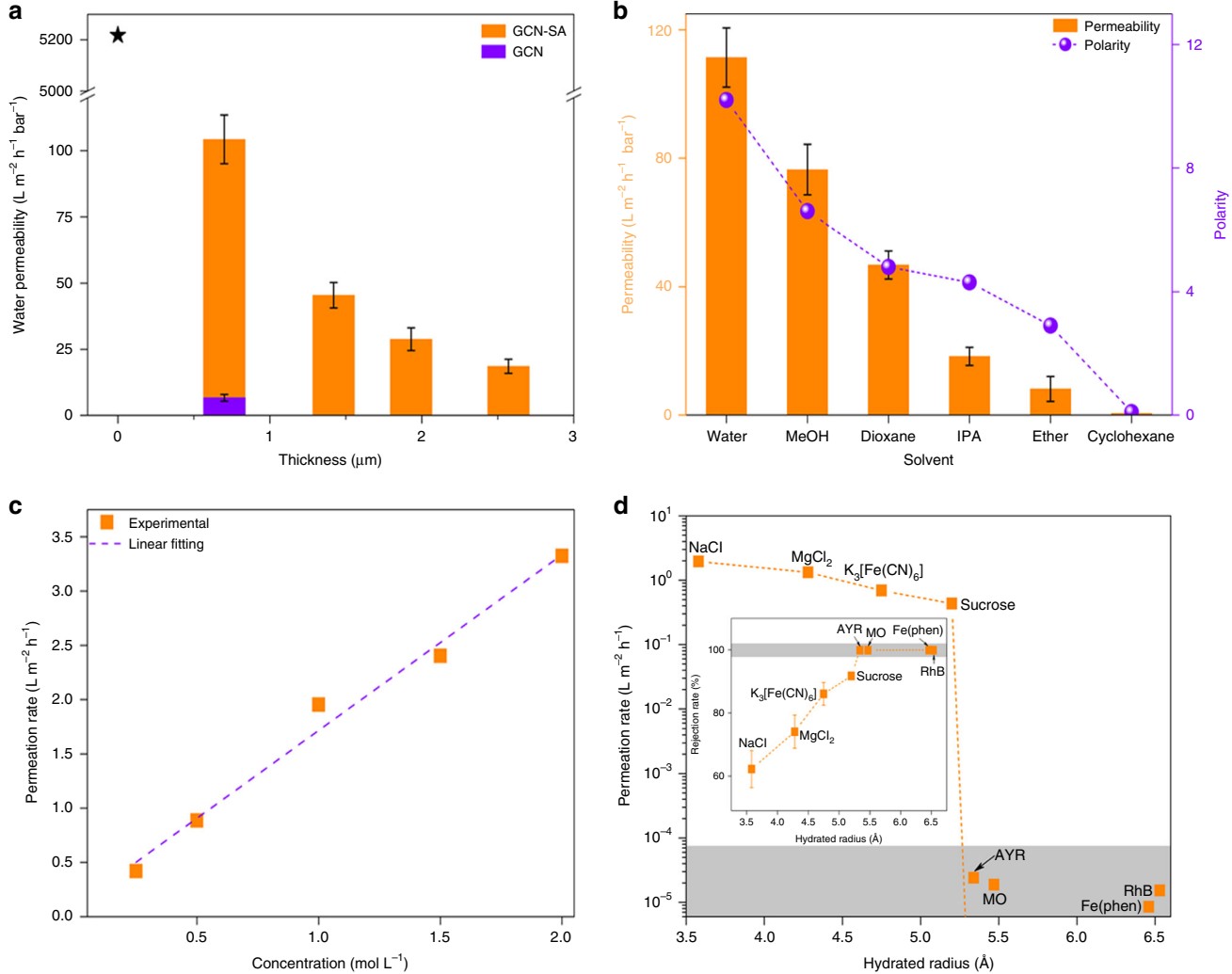

**Fig. 3** Solvent permeability and sieving performance of GCN-SA membranes. The permeability, rejection rates and permeation rates are calculated according to Eqs. 1, 2 and 3, respectively. **a** Thickness-dependent water permeability of GCN-SA membranes fabricated by varying the volume of GCN-SA nanosheet suspension. The violet column shows the water permeability over GCN membrane fabricated by vacuum filtration of 25 mL GCN nanosheet suspension. The black star gives the water permeability over blank MCE substrate for comparison. **b** Permeability of various solvents over 700-nm thick GCN-SA membrane against solvent polarity. The dotted violet line and violet balls denote the polarity of different solvents. **c** The permeation rate as a function of initial NaCl concentration at feed compartment using 700 nm-thick GCN-SA membrane. The dotted violet line corresponds to the linear fitting result of permeation rates. **d** Sieving performance of varied solutes through 700 nm-thick GCN-SA membrane. Inset: corresponding rejection rates against hydrated radii of these solutes. The dotted orange lines denote the permeation rates and rejection rates (inset) of different solutes as a function of their hydrated radii. All the error bars represent the standard deviation from three experimental data. (Fe(phen) [Fe(phen)$_3$]Cl$_2$, AYR alizarine yellow R, MO methyl orange, RhB rhodamine B)

test. A long permeation time of 12 h was set to eliminate the adsorption effect on membrane surface over permeation test. We firstly tested NaCl permeation in aqueous solution and observed the linear dependence of permeation rate on its initial concentration, suggesting ideal permeate function of GCN-SA membrane (Fig. 3c). Subsequently, various solutes with different molecular/ionic sizes, charges and shapes in aqueous solution were investigated. Plot of the permeation rates against hydrated radius of various solutes is given in Fig. 3d. A cliff drop of permeation rate occurs when the solute radius increases from 5.2 to 5.4 Å, agreeing with the d value of 10.8 Å obtained from XRD data. Species sizes larger than the d value are sieved out during permeation test. Being different from the water-mediated GO membrane where the permeation rates are independent on ion charge, the permeation rates in GCN-SA membrane are closely associated with the charge of solutes owing to the complicated electrostatic interactions from cation-SO$_4^{2-}$ and

anion-protonated GCN sheets. As shown in Fig. 3d, Na$^+$, Mg$^{2+}$, [Fe(CN)$_6$]$^{3-}$ ions with sizes smaller than d value but increasing charge magnitude exhibit decreased permeation rates in sequence. The above results reveal that anion intercalation into the protonated GCN is effective to control the interlayer spacing of GCN-based membrane for sieving at molecular precision.

**Enantioselective permeation of GCN-CSA membrane**. We further adopted enantiopure CSA as mediator instead of sulfuric acid in order to introduce chiral functionality into the membrane. The GCN-CSA composite was prepared via water-assistant ball milling method ("Methods" section) and analyzed in more details (Supplementary Note 4, Supplementary Figs. 13–17, and Supplementary Table 6). The same preparation setup and procedure applied for GCN-SA membrane were employed for GCN-CSA membrane studies. In GCN-CSA composite, the sulfonate and

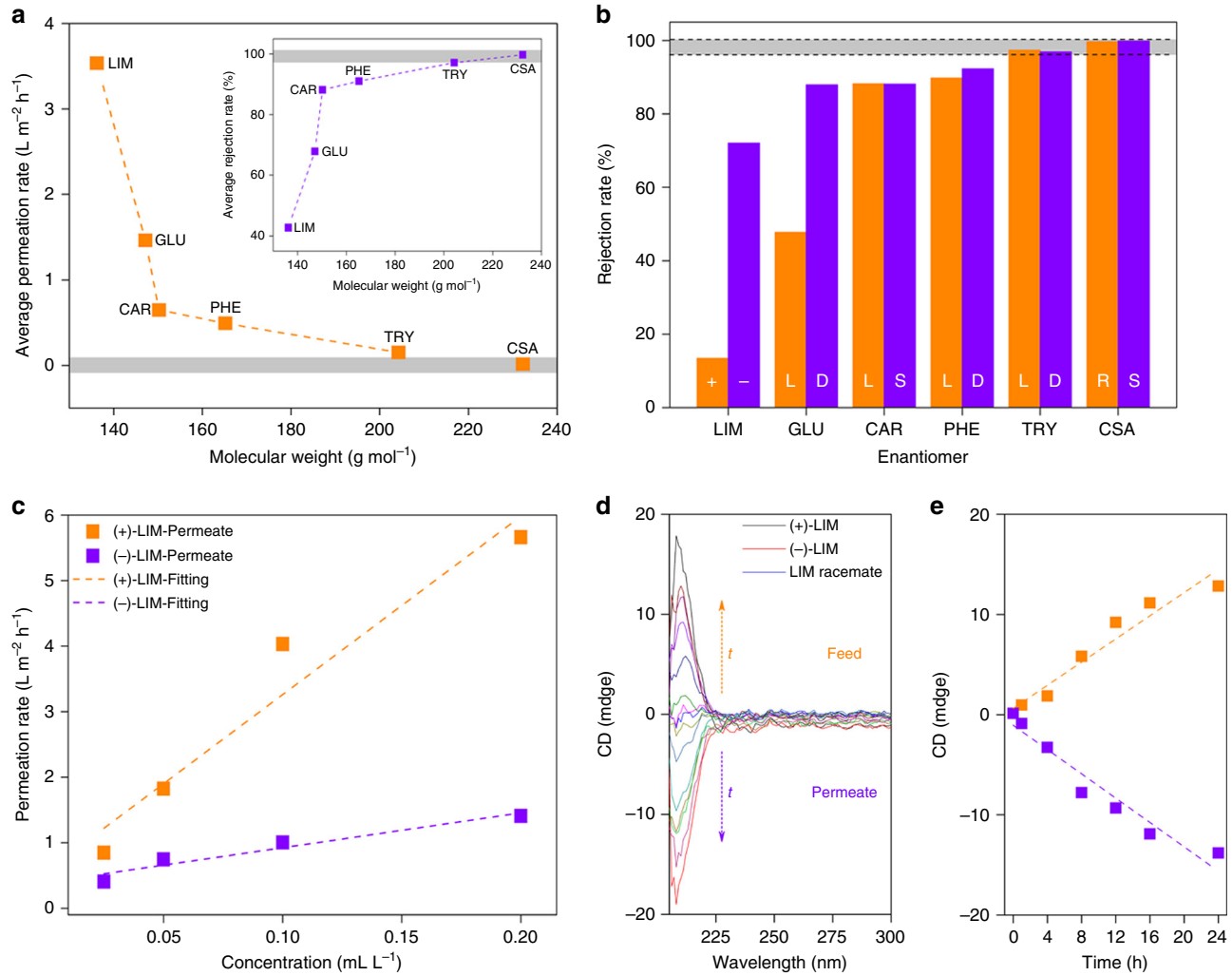

**Fig. 4** Enantioselective permeation through GCN-CSA membranes. **a**, **b** Average permeation/rejection rates (**a**) and rejection rates (**b**) of various enantiomers with increasing molecular weight (denoted by dotted orange and violet lines, respectively). Average permeation rates and average rejection rates are calculated according to Supplementary Eqs. 1 and 2, respectively. **c** The permeation rate as a function of initial (+)/(−)-LIM concentration at feed compartments. Permeation rates are calculated according to Eq. 3. The dotted orange and violet lines are linear fitting results of permeation rates of (+)-LIM and (−)-LIM, respectively. **d** CD spectra over permeation time at feed (dotted orange line) and permeate (dotted violet line) compartments using LIM racemate. The LIM racemate is a mixture of (+)-LIM and (−)-LIM with equal concentration of 0.1 mL/L (in ethanol). **e** CD signal intensity of solutions at feed and permeate compartments as a function of permeation time. The dotted orange and violet lines denote the linear fitting results of CD signal intensity of solutions at feed and permeate compartments, respectively. LIM limonene, GLU glutamic acid, CAR carvone, PHE phenylalanine, TRY tryptophan, CSA camphorsulfonic acid. All permeation tests are completed using 600-nm thick GCN-CSA membranes

organic group interacted with positively charged GCN layers via electrostatic and Van der Waals forces, respectively. Unlike GCN-SA membrane with unambiguous crystallinity, GCN-CSA membrane was prone to be amorphous (Supplementary Note 5, and Supplementary Fig. 18). High quality of the membrane was confirmed by SEM images, where no defect was observed (Supplementary Fig. 19). We selected a 600 nm-thick membrane (Supplementary Note 5, and Supplementary Fig. 20) deposited on PTFE substrate for permeation study using various chiral enantiomers for a period of 12 h (Fig. 4, Supplementary Note 6, Supplementary Fig. 21, and Supplementary Tables 3 and 4). It is difficult to identify the cutoff radii over GCN-CSA membrane. Nevertheless, the average permeation rates of enantiomers dramatically decrease from limonene to carvone as shown in Fig. 4a, which suggests a molecular weight cutoff around 150 (Supplementary Note 7, and Supplementary Figs. 22, 23).

The unique property of GCN-CSA membrane relies on the chiral environment in interlayer space derived from enantiopure

CSA ions located in-between protonated GCN sheets. The enantioselective permeation was evaluated over a series of enantiomers ranging from less polar limonene, carvone, to polar α-amino acids and CSA in various solvents owing to their different solubility (Fig. 4a). Among the species tested, permeation rates of carvone, phenylalanine, and tryptophan enantiomers are low and no enantioselective permeation is observed. Note that both R/S-CSA enantiomers are almost 100% rejected from accessing to the interlayer spacing supported by itself in GCN-CSA membrane, which is ascribed to strong hydrogen bonding of CSA in aqueous solution (Fig. 4b). Similarly, $K^+$ ion cannot enter the spacing in $K^+$ intercalated GO membrane owing to the hydration of $K^+$ ion[11]. Smaller molecules, such as limonene and glutamic acid can pass through GCN-CSA membrane with enantioselective permeation. The permeation rates of both limonene and glutamic acid enantiomers are proportional to their initial concentrations, indicating ideal permeation behavior (Fig. 4c and Supplementary Figs. 24–26). The results reveal that

incorporating chiral CSA anion into protonated GCN layers is reliable to achieve enantioselective permeation for chiral resolution.

**Enantioselective permeation mechanism**. The enantioselective permeation effect over porous membrane arises from the different adsorption and diffusion process of enantiomers, involving complicated interactions including hydrogen bonding, electrostatic and Van der Waals forces and steric-hindrance effect, etc., rather than simple size sieving effect[26]. Here, the effect of (+)/(−)-limonene adsorption on enantioselective activity of GCN-CSA material was evaluated to explore the enantioselective permeation mechanism (Supplementary Note 8, and Supplementary Fig. 27). The results indicate that GCN shows no adsorption towards both (+)-limonene and (−)-limonene, as there is no obvious change between the adsorption of the supernatant after adsorption and initial solutions (Supplementary Fig. 27a, b). However, the intensity of UV–Vis spectra of (+)/(−)-limonene supernatant decrease in comparison with their initial solution, revealing that both (+)/(−)-limonene can be adsorbed by GCN-CSA, as shown in Figs. S27c, d. The bigger absorbance difference between the supernatant and initial (+)-limonene solution suggests that GCN-CSA prefers to adsorb (+)-limonene in comparison with that of (−)-limonene. The results are in accordance with the preferred permeation of (+)-limonene over GCN-CSA membrane. Generally, a permeation behavior is associated with a kinetic adsorption-diffusion-desorption process, involving the complicated interaction among solvents, solutes and permeation medium. Therefore, it is difficult to find out a universal explanation to address the enantioselective permeation mechanism. In this work, we ascribe the selective permeation of (+)-limonene over (−)-limonene to their steric effect and thus different interaction with chiral component of CSA in GCN-CSA membrane, which in turn contributes to the different adsorption behavior as described above and final enantioselective permeation.

**Enantioselective permeation efficiency**. We also evaluated the effects of membrane thickness and loading amount of CSA on the enantioselective permeation efficiency of (+)/(−)-limonene. It was found that the permeation rates of both (+)-limonene and (−)-limonene decreased monotonously with increasing membrane thickness (Supplementary Fig. 28). Difference of permeation rates between (+)-limonene and (−)-limonene was increased when the membrane thickness increased from 0.3 to 0.6 μm, indicating enhanced separation efficiency. While further increasing thickness to 0.9 and 1.2 μm gives rise to lower separation efficiency owing to the longer path for enantiomers to transport in membrane, which exerts more evident impact on permeation-preferable (+)-limonene (Supplementary Fig. 28e). In this work, optimized permeation rates and enantioselective permeation efficiency of (+)/(−)-limonene are obtained using GCN-CSA membrane with a thickness of 600 nm. Note that the initial mass ratio of GCN to CSA is 1:5, which is actually optimized ratio for protonation and intercalation of GCN. Further increasing the starting amount of CSA does not obviously increase the loading amount of CSA in GCN, as indicated by elemental analyses shown in Supplementary Table 7. Accordingly, the efficiency of enantioselective permeation shows no improvement when using higher initial mass ratio of GCN to CSA (1:10).

**GCN functionalization with other chiral intercalators**. Furthermore, various chiral organic acids with different sizes and acidity, including 3-(2-Naphthyl)-D-alanine (NDA), (+)-camphoric acid (+CAM), and tauroursodeoxycholic acid dihydrate (TAD) have also been selected to tailor the interlayer environment of GCN (Supplementary Methods, and Supplementary Fig. 29). Specifically, the use of NDA with conjugated naphthyl structural unit was anticipated to induce non-covalent π–π stacking between NDA and GCN[27], while the intercalation turned out to be unsuccessful due to the weak acidity of NDA that fails to protonate and functionalize GCN. So was the case in +CAM, which contains two carboxyl groups. Being similar with GCN-CSA, we found that the use of sulfonic acids with stronger acidity favors the protonation and intercalation. In this regard, TAD was selected as suitable candidate to functionalize GCN, the as-prepared membrane did show enantioselective permeation capability towards, for example, (+)/(−)-limonene and L/D-penicillamine, but with moderate performance (Supplementary Fig. 30). The membrane also blocks up the permeation of larger chiral enantiomers, such as CSA and TAD. The lower efficiency is ascribed to the lower loading amount of TAD with stronger steric hindrance in-between GCN layer (Supplementary Table 8). Nevertheless, the combined results indicate the universal strategy for GCN functionalization with assistance of either inorganic or organic acids, but the exploration of other appropriate chiral acids for highly efficient enantioselective permeation is desired in future work.

**Limonene racemate separation**. We further evaluated the practical separation performance using the limonene racemate comprised of equivalent (+)/(−)-limonene as solute in ethanol under isobaric condition (Supplementary Note 9). The permeation rates of both limonene enantiomers are in linear relationship with initial feed concentrations (Fig. 4c) and the limonene racemate shows no CD signal (Fig. 4d). It is evident that CD signals in both feed and permeate compartments increase over time with opposite trend (Fig. 4d, e), indicating that (+)-limonene permeates much faster than (−)-limonene in the racemates. The preferential permeation of (+)-limonene is consistent with the permeation test using enantiopure limonene (Fig. 4c). Over a permeation period of 24 h, the ee values at both sides reach up to 89% (calculated according to Fig. 4d and Supplementary Fig. 31), showing a promising prospect for practical application. When the permeation of limonene racemate solution (5 mL) is conducted under reduced pressure using peristaltic pump, the ee value is lowered from 89 to 75% (Supplementary Note 9, and Supplementary Fig. 32), however, the operation time is decreased from 24 h to 3 min, which is believed to be more suitable for practical applications.

In summary, we achieve to control the interlayer spacing and chirality of GCN-based membranes and the stable membranes exhibit effective selective permeation effect in varied environments. The factors affecting permeation performance, especially for enantioselective permeation are complicated; hence call for further experimental and theoretical work for understanding in-depth. However, these multiple factors also provide a big space for optimizing separation performance. The current findings indicate that GCN-based membrane is promising for task-specified separation, which requires careful selection of acidic radicals with special sizes and functions. Moreover, our example of GCN-CSA membrane for enantioselective permeation will motivate the development of chiral membrane comprised of 2D layers, for example, in pharmacy industry.

## Methods

**Chemicals and reagents**. All reagents and solvents were purchased from commercial sources and used as received without further purification. Melamine (C$_3$H$_6$N$_6$, 99%), potassium ferricyanide (K$_3$[Fe(CN)$_6$], AR), (1R)-(−)-10-CSA (99%), (1S)-(−)-10-CSA (99%), L-phenylalanine (99%), D-phenylalanine (98%), L(+)-glutamic acid (99%), D-glutamic acid (98%), L(−)-carvone (99%),

S-(+)-carvone (97%), +CAM, 99% and TAD, 98% were purchased from Shanghai Macklin Biochemical Co. Ltd. (China). (+)-limonene (>95%), (−)-limonene (>95%) were purchased from Tokyo Chemical Industry (TCI) Co. Ltd (Japan). NDA, 98% was purchased from ARK Pharm, Inc., USA. Ethanol (AR, ≥99.7%), concentrated sulfuric acid (AR, 95–98%), sodium chloride (NaCl, AR) and magnesium chloride hexahydrate (MgCl$_2$·6H$_2$O, AR) were purchased from Sinopharm Chemical Reagent Co., Ltd. (China). Sucrose (AR), Rhodamine B (RhB, AR), methyl orange (MO, 96%), alizarin yellow R (AYR, AR), L-tryptophan (99%) and D-tryptophan (98%) were purchased from Aladdin Industrial Corporation (China). Fe(phen)$_3$Cl$_2$ was synthesized according to previous report [28].

**Characterization**. Powder X-ray diffraction (XRD) measurement was carried out on a Rigaku MiniFlex 600 X-ray diffractometer using Cu Kα radiation ($\lambda$ = 1.54178 Å). Elemental analyses (EA) were performed on a Vario Vario EL III Elemental Analyzer (Elementar Inc.). UV–Vis spectra of the solutions for permeation experiments were obtained using TU-1810 UV–Vis spectrophotometer. X-ray photoelectron spectra (XPS) were recorded on an ESCALab 250 high-performance electron spectrometer using monochromatized Al Kα radiation ($hv$ = 1486.7 eV) as the excitation source. Fourier transform infrared (FT-IR) spectra were performed on a SHIMADZU IR Affinity-1 spectrometer with KBr discs in a range from 4000 to 400 cm$^{-1}$. Field-emission scanning electron microscopy (FE-SEM) was carried out with a field-emission scanning electron microanalyzer (GeminiSEM 500). Transmission electron microscopy (TEM) was conducted using a JEL-2011 transmission electron microscope with an accelerating voltage of 100 kV. Atomic force microscopy images (AFM) were recorded using a tapping mode from an Asylum Research MFP-3D AFM. The samples for AFM measurements were prepared by placing a drop of the nanosheet dispersion onto a fresh cleaved mica with spin-coating. Contact angle tests were completed with Optical Contact Angle & interface tension meter (Kino SL200KS). Circular dichroism (CD) spectra were recorded on J-1500 CD spectrometer. Conductivity test was completed with DDS-11A conductivity meter equipped with Pt black electrode.

**Preparation of pristine GCN nanosheet**. GCN nanosheet was prepared according to our previously published procedures[29]. Typically, melamine (5 g) was loaded into a loosely covered crucible and then heated to 600 °C for 2 h at a rate of 1 °C min$^{-1}$ in a muffle furnace in air atmosphere, followed by cooling down with a ramping rate of 3 °C min$^{-1}$. Subsequently, the resultant yellow GCN powder (50 mg) was dispersed into 50 mL water, followed by continuous sonication for 15 h in an ultrasonic bath (150 W). The resultant suspension was centrifuged at 5000 rpm for 10 min to remove the residual large GCN particles. Finally, the light yellow GCN nanosheets were collected from the supernatant by centrifuging at 10,000 rpm for 5 min.

**Preparation of GCN-SA composite**. Bright yellow GCN-SA solution was obtained by dispersing GCN powder (700 mg) into concentrated sulfuric acid (10 mL), followed by heating at 100 °C for 12 h under constant stirring. After cooling down to room temperature, there was no insoluble GCN powder observed. Subsequently, GCN-SA suspension was obtained via dropwise adding the as-prepared solution (600 µL) into 5 mL deionized water under stirring. The white-beige precipitate (denoted as GCN-SA) was obtained by centrifuging at 10,000 rpm. The precipitate was washed with deionized water for three times to remove surface attached sulfuric acid. The GCN-SA powder was re-dispersed into 200 mL deionized water with sonication for 3 h, upon which well-dispersed GCN-SA suspension was obtained, as there was no precipitation observed after free standing for 6 months. In contrast, GCN nanosheet suspension with the same concentration gave obvious precipitate (Supplementary Fig. 1). In order to detect any leaching of SO$_4^{2-}$ anion from GCN-SA, we centrifuged the well-dispersed GCN-SA suspension at 10,000 rpm for 2 min and collected the supernatant. There was no detectable precipitation when adding the supernatant dropwise into BaCl$_2$ aqueous solution, indicating GCN-SA was stable and there was no SO$_4^{2-}$ anion leaching from GCN-SA.

**Preparation of GCN-CSA composite**. GCN powder (100 mg) was dispersed into the (1 R)-(−)-10-CSA aqueous solution (0.1 g mL$^{-1}$, 5 mL). The mixture was ball-milled at 500 rpm for 12 h. The resultant suspension was filtered to collect the powder, which was washed thoroughly with deionized water to remove residual CSA. The as-obtained powder was re-dispersed in deionized water (100 mL) via gentle sonication for several minutes. The suspension was then centrifuged at 8000 rpm for 5 min to remove large aggregated particles. Gray powder (denoted as GCN-CSA) was obtained by centrifuging the milky-like supernatant at 12,000 for 10 min.

**Preparation of membranes**. The conventional vacuum filtration method was employed to prepare the membranes using the water suspensions of GCN, GCN-SA, and GCN-CSA as described above. Typically, the suspension of GCN or GCN-SA with a concentration of 10 mg L$^{-1}$ was filtered under vacuum onto the MCE or PTFE substrate (MCE, PTFE). Both substrates are featured with average pore size of 200 nm, porosity of ~50% and a diameter of 50 mm. The thickness of these membranes can be reasonably tuned by changing the volume of suspension, which was set to be 25 mL for solvent flux and permeation tests. The as-prepared

membranes were subjected to vacuum drying for 24 h at room temperature. Although the as-prepared membranes can be peeled off, the membranes deposited on substrate were directly employed in this work. Because MCE and PTFE substrates exert little impact on permeation performance, but greatly improve the mechanical stability of the membranes. The above-described procedures were also employed for the preparation of GCN-CSA membranes using PTFE as substrate.

**Permeability and permeation tests**. The solvent permeability tests were carried out using a home-made device, as shown in Supplementary Fig. 8. The reservoir connected to feed compartment was filled with test solvent with certain volume and the permeate compartment was connected to air pump. The negative pressure inside permeate compartment was approximated to be zero. The applied supporting substrate depends on the solvent (water permeability using MCE and organic solvents using PTFE). Specifically, two glass tubes (inner diameter: 0.7 cm) prescribed as feed and permeate compartments were separated by the membrane (diameter: 2 cm) on substrate. The membrane was glued onto a silicone pad with an opening of 1 cm, which was then clamped between two O-rings and fixed to provide a leak-free environment for solvent permeability and permeation tests. Note that the membranes employed for permeation had an effective diameter of 1 cm rather than 2 cm. Supplementary Fig. 8b shows the U-shaped device for permeation tests, which is the same with component 2 in Supplementary Note 8a. The feed and permeate compartments were filled with test solution and corresponding blank solvent (typically 5 mL in each compartment), respectively. In this fashion, the isobaric permeation can be achieved and the concentration difference is considered as only driving force for permeation.

**Quantitative analyses for permeability and permeation tests**. In solvent permeability tests, the permeability of water or organic solvents was calculated using the following equation [12]:

$$\text{Permeability} = \Delta V/(A \times \Delta t \times \Delta P) \quad (1)$$

where $\Delta V$ is the permeated volume of the solvent within the duration time $\Delta t$, $A$ is the effective membrane area, $\Delta V$ can be calculated using the varied mass in sealed bottle (component 3 in Supplementary Fig. 8a). The driving force for solvent permeability is the pressure difference ($\Delta P$) calculated by the equation $\Delta P = P_1 - P_2$ ($P_1$ is determined as 1 bar induced by air pump in this work, $P_2$ is vaporizing pressure of solvent at the operating temperature). The solvent permeability tests were conducted at room temperature and the relevant vaporizing pressure of various solvents is shown in Supplementary Table 2.

In permeation tests, the solutions in feed and permeate compartments were quantitatively analyzed for calculation of rejection rates and permeation rates, after a permeation period of 12 h. Depending on the properties of solutes, varied techniques were applied to determine the solute concentration. The rejection rates ($R$) were calculated using the equation listed as follows [30]:

$$R\,(\%) = \left(1 - \frac{C_P}{C_F}\right) \times 100\% \quad (2)$$

where $C_P$ is solute concentration in permeate compartment at a permeation time of 12 h, and $C_F$ is the solute concentration at feed compartment. The permeation rates ($P$) of studied solutes were estimated by the following equation:

$$P = \frac{(C_P \times V)/(A \times \Delta t)}{\Delta C} \quad (3)$$

where $C_P$ is the varied concentration at permeate compartment, $V$ is the volume of solution at permeate compartment, $\Delta C$ is the average concentration difference between feed and permeate compartments at $t = 0$ h and 12 h (the period of permeation test is set to be 12 h). Ideally, the permeation rate should be calculated on a real-time concentration difference, as the concentration difference changes with permeation process. However, in practice we usually set the initial concentration of solutes and will not artificially intervene in the concentration in both feed and permeate sides during the permeation process. Therefore, we calculated the average permeation rate over the whole permeation period according to Eq. 3.

We repeated the permeation tests for three times to guarantee the reliability of permeation data. The permeated amounts of sodium and magnesium salts through the membranes were determined by testing the concentration of solutions in permeate compartment, which can be evaluated via conductivity test. For solute sucrose, the total organic carbon (TOC) analysis was applied for quantitative analyses. For other solutes, UV–Vis absorption spectrometer was employed to check the concentration of permeate compartment. The solute solutions are prepared with different concentrations according to their solubility in water or ethanol, as listed in Supplementary Table 3.

**Hydrated radii of solutes for permeation tests**. The hydrated radii of some molecules that can be found in previous reports, including K$^+$, Cl$^-$, Na$^+$, Mg$^{2+}$, Fe(CN)$_6^{3-}$, and sucrose were used[6]. For other molecules, we introduced the Connolly accessible area (CAA) for further calculation, which was described by the locus of the center of the solvent molecule (which is considered as a sphere) as it

rolls over the van der Waals surface of probe molecules[31]. CAA was calculated using Chem 3D Ultra Software (8.0.3 version, Cambridge-Soft, MA, USA). Energy minimization with MM2 method was performed to generate the structure of well-studied molecules, including glycerol, dextrose, $Fe(CN)_6^{3-}$, sucrose, lactose, raffinose and Ru(II)[6,32,33]. The equivalent spherical radius (CAA radius) of molecule was derived from CAA and then plotted as a function of hydrated radius, which is shown in Supplementary Fig. 9. The favorable linear relationship indicates acceptable calculation results, upon which the hydrated radii of other molecules are obtained and shown in Supplementary Table 4. For the enantiomers used for permeation tests over GCN-CSA membranes, the hydrated radii are also calculated and listed in Supplementary Table 4.

## Data availability

All data generated or analyzed during this study are included in this article and its Supplementary Information files, other data that support the findings of this study are available from the corresponding author upon request.

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

## Acknowledgements

We acknowledge support from Hefei National Laboratory for Physical Sciences at the Microscale, Hefei Science Center of Chinese Academy of Sciences, Fujian Institute of Innovation of Chinese Academy of Sciences, the National Natural Science Foundation of China (NSFC, 21571167, 51502282), the Fundamental Research Funds for the Central Universities (WK2060190053) and Anhui Province Natural Science Foundation (1608085MB28).

## Author contributions

B.L. conceived and designed the experiments. Yang W. synthesized samples, prepared the membranes, tested permeation experiments and analyzed data. N.W. helped in the synthesis of GCN-SA samples and conducted contact angle test. Yan W. and H.M. contributed to UV–Vis absorption and XPS tests. J.Z., L.X., and M.A. helped in microstructural observation and useful analyses. B.L. wrote the manuscript with input from all authors. All authors discussed the results and commented on the manuscript.

## Additional information

**Competing interests:** The authors declare no competing interests.

