## [Peer Review File · Nature Communications]

Reviewers' comments:

Reviewer #1 (Remarks to the Author):

In this manuscript, Wang et. al reported the preparation of graphite phase carbon nitride based membrane (GCN-SA and GCN-CSA) for achiral/chiral separation. (1R)-(-)-10-camphorsulfonic acid (CSA) was used as the chiral intercalator for chiral separation of the limonene isomers. Although the enantioselective permeation was evaluated over a series of enantiomers, the use of chiral intercalators should not be limited to CSA since the interlayer spacing of GCN can be precisely controlled by the selection of diverse chiral intercalators. In my opinion, diverse chiral intercalators of different sizes should be considered for chiral separation of the isomers of different sizes. What's more, the mechanisms of chiral resolution were not given. Therefore, I do not recommend the acceptance of this manuscript.

Minor points:

1. Fig. 2d, please explain why the (002) diffraction peak of GCN disappear in GCN-SA membrane.
2. As the authors stated, the hydrophilic nature of GCN-SA is due to the protonation of GCN. However, the discussion on the amphipathic property of GCN-SA is missing.
3. The CD signal of GCN-CSA should be given for a better comparison.

Reviewer #2 (Remarks to the Author):

In this work, the authors described anion intercalation in protonated GCN that tunes the interlayer spacing and the functions of GCN-based membranes for achiral/chiral separation in aqueous/organic solution. Anion/ion intercalation to control the interlayer spacing is a common method in two dimensional (2D) membrane fields, but functions of GCN-based membrane for achiral/chiral separation is limited. The manuscript presents the successful example of 2D GCN-based membrane for chiral separation. But some characterizations and content arrangement of this manuscript require further clarification, or proof, for the benefit of the readers. A revision is recommended as followings:

1. Does gentle sonication bring about the smaller lateral size of GCN-SA than those of GCN shown in AFM?

2. The thickness results of GCN-SA and GCN-CSA nanosheets by AFM are failed to get the phenomenon of exfoliation especially in Fig. S8b, 8c, the apparent platform in height profiles does not appear.
3. From the cross-sectional images of GCN-based membrane in Fig. 2 and Fig. S14, I cannot see obviously laminated structure.
4. The XRD result of GCN-CSA membrane with the PTFE support was not prone to be amorphous, this result had to be further characterized without support or other characterizations.
5. How does the GCN-SA membrane with the d value of 10 Å obtained by XRD to get the sharp cut-off between the solute radius from 5.2 to 5.4 Å in Fig. 3d?
6. The thickness of GCN-CSA membrane for achiral/chiral separation is 600 nm, while those only for size sieving are 700 nm, does the membrane thickness affect the chiral separation?
7. From the FTIR spectra of Fig. S6 and S11, there are no apparent shifts for the characteristic peaks of the functionalized GCN nanosheets, thus this result cannot prove the successful functionalization of GCN nanosheets.
8. The chiral separation mechanism of GCN-CSA membrane needs to be clarified, e.g. hydrogen bonding, electrostatic and Van der Waals forces, steric-hindrance effect, which effect domains this mechanism.
9. This paper content needs to coordinate with achiral/chiral separation of paper title, because many examples about size sieving are not on this topic.
10. The content arrangement in supplementary information should consider rearrangement to reduce reader's confusion, maybe on the basis of the appearance order in the main text.

Reviewer #3 (Remarks to the Author):

The authors prepared graphite phase carbon nitride (GCN)-based graphene-like two-dimensional membranes. The membranes had a crystalline and amphipathic structures by intercalating sulfate anion and showed not only molecular sieving performance due to its accessible spacing of 10.8 Å but also high stability in water and solvent permeation. By incorporating (1R)-(-)-10-camphorsulfonic anion (CSA), a GCN-CSA chiral membrane was prepared, which could cause enantioselective permeation separate of limonene and glutamic acid.

The performance of the prepared GCN-based membranes seems to be good. I think the methodology of the two-dimensional membrane preparation is novel and the results are interesting and of significance, if they are true.

However, evaluation and analysis method of measured data may have some uncertainty for both the liquid permeation rate and solute permeation rate. The definition and meaning of permeation rate should be understood more carefully because these values are so significant in this manuscript to decide the valuableness of the membrane performance.

In addition, especially for the chiral membrane performance, the separation mechanisms should be more clearly explained. Even if some parts of the two-dimensional layers are blocked by incorporated CSA, solute can steer around it where diffusion resistance is lower. The discussion of CSA content in the membrane seems to be very important to express enantioselective permeation, but we can see little descriptions for it. This paper may need essential improvements for publication. The following are my comments.

1. Lines 96-135, Fig.3, and p.28, Fig. S16 in SUPPLEMENTARY INFORMATION (SI);

1) Evaluation of water and solvent flux

In Fig. 3a and c (this is typo in Fig. 3, it should be “b”), the authors showed water and solvent “flux” in the unit of L/(m² h bar). This is not a flux but a permeability normalized by pressure difference as a driving force for permeation. It’s only a problem of a word, but I’m not sure how the authors calculated the permeability. According to the schematic image of the permeation experimental apparatus in Fig. S16, the permeate side seems to be evacuated by an air pump. In this case, this is not a filtration but a pervaporation (PV) measurement. So, the pressure difference is not a 1 bar but it is a vaporizing pressure that depends on temperature and solvent species. I’m not sure the permeability was correctly calculated or not.

2) Evaluation of solute permeation rate;

A permeation (diffusion) rate depends on a driving force. In this case, the driving force for solute permeation is solute concentration difference between feed side (left side of U-shaped device) and permeate side (right side of U-shaped device). True solute permeation property should be discussed on permeability, which is correctly calculated from permeation flux divided by concentration difference (C_F(t)-C_P(t)) as a driving force. The authors emphasized that permeation rate of AYR, MO, Fe(phen), and RhB were much lower than that of smaller solute molecules such as NaCl, MgCl₂, and sucrose. However, as summarized in Table S3, the initial concentration of AYR, MO, Fe(phen), and RhB are very low. The linear relationship between permeation rate and concentration at higher concentration (Fig. 3b) around 1 mol/L should not be used for explanation of adequacy of permeation rate calculation of much lower concentration measurements. The explanation in Fig.3, that “The permeation rates are normalized per 1M aqueous solutions at feed compartments.” is vague. The concentration used for the normalization should not be initial concentration. Permeation rate should be normalized by average concentration difference during the period of quasi-steady state permeation measurements. If time course data of solute concentration of both the feed and permeate side along with time are shown in SI, it is very preferable for readers to confirm the

adequacy of solute permeation rate. In addition, we can see no adsorption data of these solutes, so the reader might suspect that 12 hours is too short for eliminating the adsorption effect and that 50 mg/L is small for saturated adsorption.

3) Fig. S19, S24, S25

The authors measured the concentration of solute by UV-Vis absorption spectra. Figs. S19c, d are seems to be reasonable for evaluating the concentrations from the peak height of the spectra. However, for example, the peak locations depend on the concentration in S19a, b, and in Fig. S19f, the peak is not determined at lower concentrations. The same situations are observed for Figs. S24 (peak shift) and S25 (disappearance of peak). The authors are requested to explain in detail how to decide the value of the solute concentrations from those spectra data. This issue is critical because enantioselective permeation was detected only for the case of LIM and GLU which corresponded to the data in Figs. S19a, b, f, S24, and S25.

2. Lines 144 and 154;

The GCN-CSA membrane was amorphous, but the membrane showed the molecular weight cut-off around 150. The effective pores for selective permeation of this type of two-dimensional membrane are interlayer space of nanosheets. If it has amorphous structure, I wonder why it can show molecular sieving performance.

3. Fig. 4 and Fig. S22;

As for the solute permeation rate, there seems to be the same concern as Fig. 3.

4. Lines 149-151;

As authors pointed out, I agree that there are indeed so many factors for enantioselective permeation such as hydrogen bonding, electrostatic and Van der Waals forces, and steric-hindrance effect, etc. However, adsorption and desorption to enantioselective CSA might be the dominant factor for it. Therefore, in order to confirm the enantioselective activity of GCN-CSA material, adsorption data of LIM or GLU for bulk GCN and GCN-CSA samples are welcome. Since high enough loading of CSA in GCN would be required for expressing enantioselective permeation, the effect of CSA content on the selectivity is also better to be examined and presented.

5. Typos noticed;

Line 36; staking  stacking

Fig. 3; c  b, b  c

Fig. S19; e f, f e

Bo Liu, Ph. D., Professor
Department of Chemistry
University of Science & Technology of China (USTC)
96 Jinzhai Road, Hefei, Anhui 230026, P.R. China
Tel/ Fax: 86-551-63601123
Email: liuchem@ustc.edu.cn

Point-by-point responses to the reviewers' comments

(Reviewers' comments and the response are displayed in black and blue, respectively)

Reviewer #1 (Remarks to the Author):

In this manuscript, Wang et. al reported the preparation of graphite phase carbon nitride based membrane (GCN-SA and GCN-CSA) for achiral/chiral separation. (1R)-(-)-10-camphorsulfonic acid (CSA) was used as the chiral intercalator for chiral separation of the limonene isomers. Although the enantioselective permeation was evaluated over a series of enantiomers, the use of chiral intercalators should not be limited to CSA since the interlayer spacing of GCN can be precisely controlled by the selection of diverse chiral intercalators. In my opinion, diverse chiral intercalators of different sizes should be considered for chiral separation of the isomers of different sizes. What's more, the mechanisms of chiral resolution were not given. Therefore, I do not recommend the acceptance of this manuscript.

Response: Thanks a lot for your kind comments. In this work, we direct our research focus on the reasonable selection of inorganic and organic acids to functionalize the chemically inert GCN via protonation of GCN and introduce anions into interlayer space via electrostatic interaction, aiming to tune the distance and chemical environment of the interlayer space in the composite for selective permeation. We demonstrated the examples of SA and CSA for achiral and chiral modification of GCN, which displayed superior selective permeation over solutes with different sizes and chirality, respectively. Actually, we have examined much more inorganic and organic acids for the target. Unfortunately, their selective permeation performance is not as good as SA and CSA.

When it comes to the use of different chiral intercalators, various chiral organic acids including 3-(2-Naphthyl)-D-alanine ($C_{13}H_{13}NO_2$, abbreviated as NDA, 98%, ARK Pharm, Inc., USA), (+)-camphoric acid ($C_{10}H_{16}O_4$, abbreviated as +CAM, 99%, Shanghai Macklin Biochemical Co. Ltd., China) and tauroursodeoxycholic acid dihydrate ($C_{26}H_{45}NO_6S \cdot 2H_2O$, abbreviated as TAD, 98%, Shanghai Macklin Biochemical Co. Ltd., China) have been selected and tested. Their molecular structures are displayed in Fig. R1. Note that previous reports on interlayer intercalation of graphene and graphene oxide demonstrated that conjugated intercalators would facilitate this process via non-covalent interaction (*Chem.*

Rev., 2016, 116, 5464-5519), that is why we chose NDA with conjugated naphthyl structural unit to induce π - π stacking between NDA and GCN (Fig. R1a). However, it turned out to be unsuccessful in terms of intercalating into GCN interlayer, we ascribe this to the weak acidity of NDA or formation of inner salt that fails to protonate and functionalize GCN despite of the existence of conjugated component. So was the case in +CAM, which contains two carboxyl groups (Fig. R1b).

Fig. R1 | Molecular structures of different chiral intercalators. a, NDA. b, +CAM. c, TAD.

Actually, sulfonic acids are of stronger acidity in a diverse class of organic acids, which motivated us to prepare GCN-CSA composite using CSA that contains sulfonate. It was found that CSA can protonate GCN and further intercalate into the interlayer, following similar functionalization mechanism with SA. The GCN-CSA membrane shows highly enantioselective separation capability. We also selected TAD with sulfonate to tailor the interlayer spacing and chemical environment of GCN (Fig. R1c). However, the ball-milling assisted sonication (the same procedure for GCN-CSA) gave rise to low intercalation efficiency in GCN-TAD, probably due to its stronger steric hindrance of TAD than that of CSA. We adopt a modified experimental procedure that started from the use of acidified GCN as raw material (abbreviated as AGCN, see experimental procedures below). The as-prepared AGCN-TAD membrane does show enantioselective permeation towards, for example, (+)/(-)-limonene and L/D-penicillamine, but with moderate performance (Fig. R2). The membrane also blocks up the permeation of larger chiral enantiomers, such as CSA and TAD. We ascribe this to the lower loading amount of TAD in-between AGCN interlayer, as evidenced by the elemental analyses of AGCN-TAD and GCN-CSA (Table R1).

Fig. R2 | UV-Vis absorption spectra of initial solutions and solutions at permeate compartments after permeation. **a**, Initial feed solutions: 0.2 mL L^{-1} (+)/(-)-LIM. **b**, Initial feed solutions: 0.1 g L^{-1} L/D-PEN (PEN: penicillamine).

Table R1 Elemental analyses of GCN-CSA and AGCN-TAD.

Sample	C (wt%)	N (wt%)	S (wt%)	C/N (molar ratio of GCN)
GCN-CSA	33.60	41.16	2.85	0.65
AGCN-TAD	34.81	40.98	1.33	0.63

Note that carbon in CSA and the carbon and nitrogen in TAD were subtracted when calculating the C/N ratio. The ratio of TAD to tri-s-triazine unit is determined to be 1:7.06, which is much lower than that in GCN-CSA sample (1:4.13), considering the different molecular weights between CSA and TAD.

It is noteworthy that GCN in chiral organic acids solution can only form suspensions with limited concentration due to larger organic groups, which differs from the case in homogeneous GCN-SA solution of high concentration. The non-homogeneous system precludes high intercalation efficiency to some extent, thereby leading to lower precision in tailoring interlayer distance of GCN. Nonetheless, the above results and discussion prove the universal strategy for GCN functionalization with assistance of either inorganic or organic acids. Our future work will focus on the use of other appropriate chiral acids for highly efficient enantioselective permeation.

Regarding to the chiral separation mechanism, please refer to the response to Reviewer 3, point 4. The revision can be also found in main text (pages 9-10) and Supplementary Section 8.3 (page 45).

Experimental Procedures for AGCN-TAD:

1) AGCN preparation: GCN-SA of 5 mL solution was heated at 70 °C for 1h, upon which 70 mL deionized water was injected and 7.5 g NH₄Cl powder was added. The suspension was subjected to stirring for 1 h and allowed to stand for another 0.5 h. And then the hot filtration was applied to obtain colorless filtrate, which was quickly transferred to ice-water bath and stirred for 1 h to obtain white suspension. Subsequently, the suspension was centrifuged at 8000 rpm for 5 min. The as-obtained precipitation was washed with H₂O and ethanol for 3 times, followed by drying at 60 °C under vacuum. Finally, the acidized GCN powder was obtained and denoted as AGCN. (*Nanoscale*, 2015, 7, 8701-8706)

2) AGCN-TAD preparation: AGCN of 10 mg was dispersed into 0.2 M hydrochloric DMF solution (10 mL) and sonicated for 2 hrs to obtain colorless solution. Excess amount of TAD (100 mg) was added into and quickly dissolved in the solution, which was then stirred for 1 h at room temperature. Anti-solvent CHCl₃ (20 mL) was selected and added into the solution, the as-obtained colloidal precipitation was washed with CHCl₃ for 3 times to remove free TAD and then re-dispersed into 10 mL H₂O, followed by sonication for 3 hrs to obtain highly dispersed AGCN-TAD without obvious precipitation after free standing for one month.

Relevant discussion and comments have been supplemented in revised main text (pages 9-10, pages 11-12) and Supplementary Section 9 (pages 49-51).

Minor points:

1. Fig. 2d, please explain why the (002) diffraction peak of GCN disappear in GCN-SA membrane.

Response: It is well acknowledged that bulk GCN prepared in air atmosphere is of low crystallinity, as indicated by the two broad diffraction peaks at ~13.1 ° (100) and 27.6 ° (002), which are assigned to the in-plane packing of tri-s-triazine motifs and interlayer stacking of conjugated aromatic rings, respectively. Successful intercalation would definitely result in change of the interlayer spacing (*d*), which can be reflected by the diffraction peak evolution. SA functionalization in homogeneous GCN-SA suspension contributed to the intercalation of sulfate ions in-between GCN layers, which changed the *d* value from 3.26 Å ($2\theta = 27.6^\circ$) to 14.06 Å ($2\theta = 6.28^\circ$) and 7.12 Å ($2\theta = 12.42^\circ$). Here, the disappearance of (002) diffraction peak accompanied with newly emerged peaks indicates the changed *d* value and hence successful intercalation of SA into GCN interspace.

2. As the authors stated, the hydrophilic nature of GCN-SA is due to the protonation of GCN. However, the discussion on the amphipathic property of GCN-SA is missing.

Response: Yes. The protonation of GCN brings about hydrophilic nature of GCN-SA, as evidenced by the contact angle change in Fig. S15 in Supplementary Information. We observe that GCN-SA membrane is permeable over solvents with a wide range of polarity, as shown in Fig. 3b in main text. The most hydrophilic liquid of water shows highest permeability and the most hydrophobic liquid of cyclohexane shows the lowest permeability. Among these, other organic solvents, such as methanol and dioxane, also show favorable solvent permeability despite of the more hydrophobic property in comparison with that of water. The combined findings support the amphipathic property of GCN-SA. Also, we experimentally find that GCN-SA is capable of highly dispersing in organic solvents, such as ethanol, methanol, IPA, etc., distinguishing it from pristine GCN that can only poorly disperse in most common solvents, upon long-time sonication (*J. Am. Chem. Soc.*, 2013, 135, 18-21). We ascribe this to the structural attributes of GCN-SA, in which the conjugated basal plane is apparently hydrophobic, while the oxygen-containing groups induced by SA functionalization can endow GCN-SA with hydrophilicity (*J. Am. Chem. Soc.*, 2017, 139, 6026-6029).

Note that our solvent permeability experiments are also in line with the well-documented slip flow theory, which implies that more hydrophobic liquid would lead to lower permeability under the same pressure, because of stronger interaction between more hydrophobic liquid and GCN. As expected by the slip flow theory, the permeability decreases with the decreasing of polarities, which agrees with the cases of carbon nanotubes and graphene (*Nature*, 2005, 438, 44; *Adv. Funct. Mater.*, 2013, 23, 3693-3700.)

The related discussion has been added into revised main text (page 6).

3. The CD signal of GCN-CSA should be given for a better comparison.

Response: The CD spectra of as-prepared GCN and GCN-CSA samples are shown in Fig. R3. For better comparison, the CD spectrum of pure aqueous CSA solution is also recorded as a reference. Only weak noise is detected for GCN, while the CD signal centered around 290 nm is observed for GCN-CSA, showing identical peak location with that of CSA. The results indicate successful functionalization and chiral attribute of GCN-CSA.

Fig. R3 | CD signal of as-prepared GCN and GCN-CSA. The CD spectrum of aqueous CSA solution with high concentration (200 mg L^{-1}) is provided as a reference.

Reviewer #2 (Remarks to the Author):

In this work, the authors described anion intercalation in protonated GCN that tunes the interlayer spacing and the functions of GCN-based membranes for achiral/chiral separation in aqueous/organic solution. Anion/ion intercalation to control the interlayer spacing is a common method in two-dimensional (2D) membrane fields, but functions of GCN-based membrane for achiral/chiral separation is limited. The manuscript presents the successful example of 2D GCN-based membrane for chiral separation. But some characterizations and content arrangement of this manuscript require further clarification, or proof, for the benefit of the readers. A revision is recommended as followings:

We thank for the valuable comments from reviewer 2.

1. Does gentle sonication bring about the smaller lateral size of GCN-SA than those of GCN shown in AFM?

Response: The gentle sonication is insufficient to change the lateral size of GCN or well disperse GCN. The smaller lateral size of GCN-SA than those of GCN is primarily attributed to the use of strong acid, which partially cut GCN into smaller piece depending on the SA concentration and sonication time. This is consistent with the result reported in literature (*ACS Nano*, 2015, 9, 12480-12487).

2. The thickness results of GCN-SA and GCN-CSA nanosheets by AFM are failed to get the

phenomenon of exfoliation especially in Fig. S8b, 8c, the apparent platform in height profiles does not appear.

Response: From AFM image of GCN (Fig. S2b), we can observe the typical monolayer feature prepared by using water as solvent under sonication. GCN can be well exfoliated due to the relatively weak interaction in-between GCN layers. In contrast, electrostatic interaction in-between GCN-SA and GCN-CSA layers make them apt to aggregating when the solvent is removed during AFM sample preparation. That is why the GCN-SA and GCN-CSA suspension is highly stable, but the particle-like morphologies are observed in AFM measurement in their dried sample. The thickness of GCN-SA and GCN-CSA range from few nanometer to about 10 nanometer as shown in Figs. S2f and Fig. S8f. Note that the observed lateral size of GCN-SA and GCN-CSA layers is around hundreds of nanometers. This phenomenon is consistent with the result reported previously (*J. Am. Chem. Soc.*, 2017, 139, 11698-11701)

TEM images show more identifiable morphology of studied samples (pristine bulk GCN is also given for comparison), as shown in Fig. R4. Bulk GCN exhibits typical platelet-like structures with compact stacking (Fig. R4a), while GCN-SA and GCN-CSA are featured with laminar texture (Figs. R4c and R4d), despite of the thicker stacking in comparison with that of GCN nanosheets suspension with low concentration prepared by long-time sonication (Fig. R4b).

Fig. R4 | TEM images of GCN-based samples. **a**, pristine bulk GCN. **b**, GCN nanosheets. **c**, GCN-SA nanosheets. **d**, GCN-CSA nanosheets.

3. From the cross-sectional images of GCN-based membrane in Fig. 2 and Fig. S14, I cannot see obviously laminated structure.

Response: As revealed in AFM images, the thickness of GCN-SA and GCN-CSA is around 10 nm due to the aggregation of monolayers via electrostatic interaction. Top-viewed and cross-sectional SEM images are recorded at micrometer scale aiming to evaluate surface uniformity (holes and cracks, smoothness, etc.) and membrane thickness. Therefore, we are not expecting to see laminated structure from SEM images for GCN-SA and GCN-CSA samples. To understand the structure and morphology of GCN-SA and GCN-CSA membrane, we need to combine the information revealed from AFM, SEM and TEM measurements together. Conclusively, pinhole- and crack-free membrane of GCN-SA and GCN-CSA are made up from plate-like particles with lateral size of ca. hundred of nanometer and thickness of ca. 10 nm. In addition, similar morphologies are also found in various GCN-based membranes in literature (*J. Membr. Sci.*, 2015, 490, 72-83; *J. Membr. Sci.*, 2015, 475, 281-289; *Appl. Catal. B: Environ.*, 2016, 194, 134-140; *Angew. Chem. Int. Ed.*, 2017, 56, 8974-8980, etc).

4. The XRD result of GCN-CSA membrane with the PTFE support was not prone to be amorphous, this result had to be further characterized without support or other characterizations.

Response: The XRD pattern of GCN-CSA membrane without any support is now shown in Fig. R5. Compared with pristine bulk GCN, the peak intensity is significantly weakened, showing a less periodic texture. The evolution of diffraction peaks clearly indicates that the as-prepared GCN-CSA membrane is more prone to be amorphous, as stated in main text. The XRD result may also serve as complementary evidence for successful exfoliation of GCN via CSA functionalization.

The corresponding result has also been added into Supplementary Section 3 (page 22).

Fig. R5 | XRD patterns of pristine bulk GCN and GCN-CSA membrane without PTFE support (XRD pattern of blank PTFE is provided for comparison).

5. How does the GCN-SA membrane with the d value of 10 Å obtained by XRD to get the sharp cut-off between the solute radius from 5.2 to 5.4 Å in Fig. 3d?

Response: As stated in main text, the intercalation of sulfate ion increases the d value of GCN-SA by ~ 10.8 Å. The permeation rate is plotted against with solutes with various hydrated radius, as shown in Figure 3d in main text. A sharp decrease (cutoff) of permeation rate is observed when the solute radius increases from 5.2 Å to 5.4 Å, which means that the hydrated diameter of solutes increases from 10.4 Å to 10.8 Å. The result keeps good consistence with the interlayer spacing value of GCN-SA of 10.8 Å.

6. The thickness of GCN-CSA membrane for achiral/chiral separation is 600 nm, while those only for size sieving are 700 nm, does the membrane thickness affect the chiral separation?

Response: As shown in Fig. 4 in main text, the GCN-CSA membrane shows favorable chiral separation effect over limonene, which is an optimized result using 600 nm thick GCN-CSA membrane. As shown in Figs. R6a-d, the permeation rates of both (+)-LIM and (-)-LIM decrease monotonously with increasing membrane thickness. Difference of permeation rates between (+)-LIM and (-)-LIM is increased when the membrane thickness increases from 0.3 μm to 0.6 μm , indicating enhanced separation efficiency. While further increasing thickness to 0.9 and 1.2 μm gives rise to lower separation efficiency owing to the longer path for enantiomers to transport in membrane, which exerts more evident impact on permeation-preferable (+)-LIM, thus leading to evidently decreased permeation rate (Fig. R6e). In our experiment, optimized permeation rates and chiral separation efficiency are obtained using GCN-CSA membrane with a thickness of 600 nm.

Fig. R6 | a-d, UV-Vis absorption spectra of solutions at permeate compartments using 0.2 mL L⁻¹ (+)-LIM or (-)-LIM as initial feed solutions and membranes with different thickness. a, 0.3 μm. b, 0.6 μm. c, 0.9 μm. d, 1.2 μm. e, Variation of permeation rates of (+)-LIM and (-)-LIM as a function of membrane thickness, the permeation rates are calculated according to Equation (R1). Inset: the dependence of $P_{(+)-LIM}/P_{(-)-LIM}$ on membrane thickness, where P represents permeation rate. Higher $P_{(+)-LIM}/P_{(-)-LIM}$ value indicates greater difference of permeation rates between (+)-LIM and (-)-LIM.

The corresponding result has also been added into main text (pages 10-11) and Supplementary Section 8.4 (page 46).

7. From the FTIR spectra of Fig. S6 and S11, there are no apparent shifts for the characteristic peaks of the functionalized GCN nanosheets, thus this result cannot prove the successful functionalization of GCN nanosheets.

Response: All these findings primarily aim to indicate the retained tri-s-triazine-based framework of GCN after protonation. In this work, the functionalization of GCN nanosheets can be complementally proved by other characterizations including XPS and elemental analyses. Actually, a series of reports on GCN nanosheets functionalized by protonation also reveal similar FT-IR results with no apparent shifts (*J. Am. Chem. Soc.*, 2009, 131, 50-51; *Small*, 2014, 10, 12, 2382-2389; *J. Mater. Chem. A*, 2013, 1, 14766-14772; *Nanoscale*, 2015, 7, 8701-8706; etc.).

Combing your comments on FT-IR spectra and previous reports, we have now modified the relevant explanation in Supplementary Information (pages 14 and 20).

8. The chiral separation mechanism of GCN-CSA membrane needs to be clarified, e.g. hydrogen bonding, electrostatic and Van der Waals forces, steric-hindrance effect, which effect domains this mechanism.

Response: please refer to the response to Reviewer 3, point 4. The revision can be also found in main text (pages 9-10) and Supplementary Section 8.3 (page 45).

9. This paper content needs to coordinate with achiral/chiral separation of paper title, because many examples about size sieving are not on this topic.

Response: Thanks for your kind suggestion. Taking your suggestion and the topic of main text into consideration, we believe that this work entitled with “Graphite-phase carbon nitride based membrane for selective permeation” would be more appropriate.

10. The content arrangement in supplementary information should consider rearrangement to reduce reader’s confusion, maybe on the basis of the appearance order in the main text.

Response: According to your advice, the content in Supplementary Information has now been rearranged based on the appearance order in the main text.

Reviewer #3 (Remarks to the Author):

The authors prepared graphite phase carbon nitride (GCN)-based graphene-like two-dimensional membranes. The membranes had a crystalline and amphipathic structures by intercalating sulfate anion and showed not only molecular sieving performance due to its accessible spacing of 10.8 Å but also high stability in water and solvent permeation. By incorporating (1R)-(-)-10-camphorsulfonic anion (CSA), a GCN-CSA chiral membrane was prepared, which could cause enantioselective permeation separate of limonene and glutamic acid.

The performance of the prepared GCN-based membranes seems to be good. I think the methodology of the two-dimensional membrane preparation is novel and the results are interesting and of significance, if they are true.

However, evaluation and analysis method of measured data may have some uncertainty for both the liquid permeation rate and solute permeation rate. The definition and meaning of permeation rate should be understood more carefully because these values are so significant in this manuscript to decide the valuableness of the membrane performance.

In addition, especially for the chiral membrane performance, the separation mechanisms

should be more clearly explained. Even if some parts of the two-dimensional layers are blocked by incorporated CSA, solute can steer around it where diffusion resistance is lower. The discussion of CSA content in the membrane seems to be very important to express enantioselective permeation, but we can see little descriptions for it. This paper may need essential improvements for publication. The following are my comments.

Response: We thank a lot for the constructive and valuable comments from reviewer 3. Please see the detailed response to the comments as follows.

1. Lines 96-135, Fig.3, and p.28, Fig. S16 in SUPPLEMENTARY INFORMATION (SI);

1) Evaluation of water and solvent flux

In Fig. 3a and c (this is typo in Fig. 3, it should be “b”), the authors showed water and solvent “flux” in the unit of L/(m² h bar). This is not a flux but a permeability normalized by pressure difference as a driving force for permeation. It’s only a problem of a word, but I’m not sure how the authors calculated the permeability. According to the schematic image of the permeation experimental apparatus in Fig. S16, the permeate side seems to be evacuated by an air pump. In this case, this is not a filtration but a pervaporation (PV) measurement. So, the pressure difference is not a 1 bar but it is a vaporizing pressure that depends on temperature and solvent species. I’m not sure the permeability was correctly calculated or not.

Response: Sorry for the confusion about flux and permeability. In previous version of manuscript, we directly adopted pressure of 1 bar generated from pump as driving force for solvent permeation rate calculation, omitting the effect from vaporizing pressure of solvents themselves. Using the set-up of solvent permeation tests as schemed in Fig. S16, the driving force for permeation rates should be the pressure difference (ΔP) calculated by the equation $\Delta P = P_1 - P_2$ (P_1 is determined as 1 bar induced by air pump in this work, P_2 is vaporizing pressure of solvent at the operating temperature). The solvent permeation tests are conducted at room temperature and the relevant vaporizing pressure of various solvents is shown in Table R2. Accordingly, the water permeability over GCN-SA membrane with different thickness, together with solvent permeability, has been corrected, which is shown in Fig. R7. The final permeation of solvents with low evaporating pressure is not dramatically changed, while the permeation rate of low boiling-point solvent like ether, increased by about two-fold. Nevertheless, the relative trend of permeation rates for various solvents is not changed.

The water permeability of blank MCE/PTFE substrates and GCN-SA membranes with different thickness, as well as solvent permeability have been corrected and highlighted in

yellow in main text (Fig. 3) and Supplementary Section 4.2 (page 28).

Table R2 Vaporizing pressure and pressure difference (ΔP) of various solvents under solvent permeation tests.

Solvent	Vaporizing pressure P_2 (kPa)	ΔP (kPa)
H ₂ O	2.338	97.662
MeOH	5.947	94.053
Dioxane	3.84	96.16
IPA	4.41	95.59
Ether	58.67	41.33
Cyclohexane	10.34	89.66

The vaporizing pressure (P_2) can be obtained from the Langer's Handbook of Chemistry.

Fig. R7 | Solvent permeability through GCN-SA membrane. a, Thickness-dependent water permeability of GCN-SA membranes. The black star gives the water permeability over blank MCE substrate for comparison. **b,** Permeability of various solvents over 700-nm thick GCN-SA membrane against solvent polarity.

2) Evaluation of solute permeation rate;

A permeation (diffusion) rate depends on a driving force. In this case, the driving force for solute permeation is solute concentration difference between feed side (left side of U-shaped device) and permeate side (right side of U-shaped device). True solute permeation property should be discussed on permeability, which is correctly calculated from

permeation flux divided by concentration difference (CF(t)-CP(t)) as a driving force. The authors emphasized that permeation rate of AYR, MO, Fe(phen), and RhB were much lower than that of smaller solute molecules such as NaCl, MgCl₂, and sucrose. However, as summarized in Table S3, the initial concentration of AYR, MO, Fe(phen), and RhB are very low. The linear relationship between permeation rate and concentration at higher concentration (Fig. 3b) around 1 mol/L should not be used for explanation of adequacy of permeation rate calculation of much lower concentration measurements.

Response: In this work, the initial concentrations are different due to different solubility of the solutes utilized. We realized that the linear relationship existing in aqueous solutions of solutes at high concentrations might not be applicable for evaluating the adequacy of permeation rate calculation of much lower concentration measurements. The permeation rates are calculated by monitoring the aqueous solution of permeate compartment and normalized per 1M aqueous solutions at feed compartments, being identical with the method provided in previous work provided by Geim et al (*Science*, **2014**, 243, 752-754).

The explanation in Fig.3, that “The permeation rates are normalized per 1M aqueous solutions at feed compartments.” is vague. The concentration used for the normalization should not be initial concentration. Permeation rate should be normalized by average concentration difference during the period of quasi-steady state permeation measurements. If time course data of solute concentration of both the feed and permeate side along with time are shown in SI, it is very preferable for readers to confirm the adequacy of solute permeation rate.

Ideally, the permeation rate should be calculated on a real-time concentration difference, as the concentration difference change with permeation process. However, in practice we usually set the initial concentration of solutes and will not artificially intervene in the concentration in both feed and permeate sides during the permeation process. Therefore, we calculated the average permeation rate according over the whole permeation period, according to the following equation:

$$P = \frac{(C_p \times V) / (A \times \Delta t)}{\Delta C} \quad (R1)$$

where C_p is the solute concentration at permeate compartment, V is the volume of solution at permeate compartment. ΔC is the average concentration difference between feed and permeate compartments at $t=0$ h and 12 hrs (the period of permeation test is set to be 12 hrs). The permeation rates of the solutes calculated based on Equation (R1) are displayed in Fig. R8a.

In order to eliminate the effect of initial concentration difference of various solutes and to provide comparable average permeation rates among different solutes, all the initial concentration of tested solutes is set at 0.005 M (solvent: water).

Obviously, after taking average concentration difference into calculation according to the above Equation (R1), permeation rates of NaCl, MgCl₂, K₃[Fe(CN)₆] and sucrose at the low initial concentration is similar with the previous results with high initial concentration (Figs. R8a and R8b). Note that no UV-Vis absorption signals of solutions in permeate compartments were observed for AYR, MO, [Fe(phen)₃]Cl₂ and RhB during the measurements lasting for even three days, keeping good consistency with our previous results. The scattered points fall in the grey area of Figs. R8a and R8b represent the detection limit. The molecular sieving effects were evaluated again via plotting the permeation rate against the hydrated solute radius; the results are shown in Fig. R8b. The data reveals the same cut-off behavior of the membrane.

Fig. R8 | Permeation of different solutes through 700-nm thick GCN-SA membranes. a, Permeation rates of all solutes with different initial concentration calculated by Equation (R1). **b,** Sieving performance of varied solutes with the same initial concentration of 0.005 M through 700 nm-thick GCN-SA membrane. The permeation rates of all solutes are also calculated by Equation (R1).

In addition, we can see no adsorption data of these solutes, so the reader might suspect that 12 hours is too short for eliminating the adsorption effect and that 50 mg/L is small for saturated adsorption.

To better illustrate the adsorption effect of studied solutes, the concentration of both feed and permeate compartments over a period of 12 hrs are listed in Table R3. For permeable solutes NaCl, MgCl₂, K₃[Fe(CN)₆] and sucrose, the initial concentrations are high enough for saturated adsorption (provided the adsorption really takes place). As shown in Table R3, the total amount of solutes from both the feed and permeate sides is very close to that in the

initial feed solutions, which indicates that the sieving performance is not falsified by adsorption effect. In this regard, we further crosschecked the possible adsorption via testing cycling performance of the membrane using fresh initial solution and pure solvent in both feed and permeate sides, respectively. The permeation performance is similar for three cycles, revealing the negligible adsorption effect.

As for the dye solutes with larger sizes, no obvious permeation was observed and negligible concentration decreasing was detected when using initial concentrations of 50 mg L⁻¹ or even much higher initial concentrations of 0.005 M. This result suggests that dyes with initial concentrations of 50 mg L⁻¹ are high enough for saturated adsorption. In addition, the feed compartments of these dye solutions exhibited no detectable UV-Vis spectra difference (that is, no detectable concentration decrease) when further prolonging time from 12 hrs to three days, which manifests 12 hrs is enough for saturated adsorption.

Table R3 Concentration of feed and permeate compartments after a permeation period of 12 hrs.

Solutes	Initial concentration	Feed	Permeate
NaCl	1 M	0.6994 M	0.2641 M
MgCl ₂	1 M	0.7507 M	0.1945 M
K ₃ [Fe(CN) ₆]	3.0373×10 ⁻³ M (1 g L ⁻¹)	2.527×10 ⁻³ M	3.523×10 ⁻⁴ M
sucrose	1 M	0.8995 M	0.0744 M
*AYR	1.7409×10 ⁻⁴ M (50 mg L ⁻¹)	1.6969×10 ⁻⁴ M	/
*MO	1.5275×10 ⁻⁴ M (50 mg L ⁻¹)	1.4958×10 ⁻⁴ M	/
*Fe(phen)	7.8442×10 ⁻⁵ M (50 mg L ⁻¹)	7.5700×10 ⁻⁵ M	/
*RhB	1.0438×10 ⁻⁴ M (50 mg L ⁻¹)	1.0115×10 ⁻⁴ M	/

* the feed concentrations of dye solutions keep unchanged after prolonging time from 12 hrs to three days.

The above discussion and modifications have been highlighted in yellow in main text (Fig. 3), Supplementary Section 4.2 (pages 28-29) and Supplementary Section 7 (pages 37-39).

3) Fig. S19, S24, S25

The authors measured the concentration of solute by UV-Vis absorption spectra. Figs. S19c, d are seems to be reasonable for evaluating the concentrations from the peak height of the spectra. However, for example, the peak locations depend on the concentration in S19a, b, and in Fig. S19f, the peak is not determined at lower concentrations. The same situations are observed for Figs. S24 (peak shift) and S25 (disappearance of peak). The authors are

requested to explain in detail how to decide the value of the solute concentrations from those spectra data. This issue is critical because enantioselective permeation was detected only for the case of LIM and GLU which corresponded to the data in Figs. S19a, b, f, S24, and S25.

Response: The peak shifts of (+)/(-)-LIM calibration curves in Figs. S19a and S19b as well as peak disappearance in Figs. S24 and S25 (marked as Figs. S25 and S26 in revised Supplementary Information, respectively) are observed. With decreasing concentrations, the peak variation is ascribed to the enhanced effect from solvent peak.

As shown in the insets of Figs. S19a, S19b and S19f, the linear dependence of absorbance on concentration is favorable ($R^2 > 0.99$), the absorption peak locations of solutes at different concentrations are determined based on the sample with highest concentration within our test range. For example, in Fig. S19a, the absorbance peak location is 211 nm for $0.2 \text{ mL}\cdot\text{L}^{-1}$ (+)-LIM, this location is also applicable for (+)-LIM with other concentrations ($0.1, 0.05, 0.025$ and $0.0125 \text{ mL}\cdot\text{L}^{-1}$) to determine the absorbance values, which is evidenced by the linear calibration curve.

The explanation is now added in the Supplementary Section 6 (pages 34-35).

2. Lines 144 and 154;

The GCN-CSA membrane was amorphous, but the membrane showed the molecular weight cut-off around 150. The effective pores for selective permeation of this type of two-dimensional membrane are interlayer space of nanosheets. If it has amorphous structure, I wonder why it can show molecular sieving performance.

Response: We understand the concerns from the reviewer. The PXRD patterns in both powder and membrane indicate that GCN-CSA structurally feature with a less periodic texture (Figs. S9 and S12). Combining the information driven from PXRD, SEM, AFM and TEM, the structure of GCN-CSA is described to be long-range disorder and short-range order. The molecular sieving behavior of GCN-CSA originates from the interlayer spacing induced by CSA intercalation within ordered region in short range.

3. Fig. 4 and Fig. S22;

As for the solute permeation rate, there seems to be the same concern as Fig. 3.

Response: Being similar with the concern in Fig. 3 that you kindly referred to, we have also

crosschecked the average permeation rates of all tested enantiomers by taking average concentration difference (as driving force) into calculation (Equation R1), which is now shown in Fig. R9a. One can see that the variation tendency keeps unchanged when compared with our previous results. Accordingly, the permeation rates of each pair of enantiomers according to the Equation (R1) (considering average concentration difference) have also been displayed in Fig. R9b.

Note that the “Average permeation rate” on y-axis in Fig. R9a is calculated according to the rates provided in Fig. R9b and the following equation:

$$\text{Average permeation rate} = (P_x + P_y) / 2 \quad (\text{R2})$$

where P_x and P_y are permeation rates (considering average concentration difference) of the two enantiomers with the same molecular weight, respectively. For example, P_x is the permeation rate of (+)-LIM and P_y is the permeation rate of (-)-LIM provided in Fig. R9b.

The main issue lies in the increased permeation rate difference between (+)-LIM and (-)-LIM when introducing concentration difference as driving force. This is reasonable because less permeable (-)-LIM is featured with higher concentration difference (ΔC in Equation R1) between feed and permeate compartments, and accordingly, more permeable (+)-LIM is featured with lower concentration difference (ΔC in Equation R1). The same variation tendency can also be found in L/D-GLU.

In addition, the permeation rates of (+)/(-)-LIM and L/D-GLU as a function of their initial concentrations are also calculated by Equation (R1), as shown in Figs. R9c and R9d. The variation tendency is also very similar with our previous results. (Here we feel sorry for the typos of horizontal axes shown in previous Fig. 4c and Fig. S23, which have now been corrected.)

The above results have been highlighted in main text (Fig. 4) and Supplementary Section 8 (pages 41-44).

Fig. R9 | Permeation of enantiomers and through 600-nm thick GCN-CSA membranes, all the permeation rates are calculated by Equation (R1). **a**, The variation of average permeation rates of different enantiomers as a function of molecular weight. Inset: average rejection of various enantiomers with increasing molecular weight. **b**, Permeation rates of each pair of enantiomers as a function of molecular weight. **c**, **d**, Permeation rates of (+)/(-)-LIM (**c**) and L/D-GLU (**d**) as a function of their initial concentrations.

4. Lines 149-151;

As authors pointed out, I agree that there are indeed so many factors for enantioselective permeation such as hydrogen bonding, electrostatic and Van der Waals forces, and steric-hindrance effect, etc. However, adsorption and desorption to enantioselective CSA might be the dominant factor for it. Therefore, in order to confirm the enantioselective activity of GCN-CSA material, adsorption data of LIM or GLU for bulk GCN and GCN-CSA samples are welcome. Since high enough loading of CSA in GCN would be required for expressing enantioselective permeation, the effect of CSA content on the selectivity is also better to be examined and presented.

Response: In order to confirm the adsorption effect over enantioselective activity of GCN-CSA material, we conducted the adsorption experiments as follows: powder samples of 100 mg (GCN and GCN-CSA) were dispersed into 20 mL 0.1 mL L⁻¹ (+)-LIM or (-)-LIM solution. We use 100 mg powder sample instead of membrane for adsorption test, because sample amount on membrane is very low so that the adsorption effect is hardly observed. After free standing for 12 hrs (the duration time is enough for adsorption-desorption equilibrium), the suspensions were centrifuged at 10000 rpm for 5min to obtain the supernatant, which were then subjected to UV-Vis absorption test. The results are shown in Fig. R10. As indicated in Figs. R10a and R10b, GCN shows no adsorption towards both (+)-LIM and (-)-LIM, as there is no obvious change between the spectra of the supernatant and initial solutions.

The intensity of UV-Vis spectra of (+)/(-)-LIM supernatant slightly decrease in comparison with their initial solution, revealing both (+)/(-)-LIM can be adsorbed by GCN-CSA as shown in Figs. R10c and R10d. The bigger absorbance difference between the supernatant and initial (+)-LIM solution suggests that GCN-CSA prefers to adsorb (+)-LIM when compared with that of (-)-LIM. The results are in accordance with the preferred permeation of (+)-LIM over (-)-LIM through GCN-CSA membrane.

Fig. R10 | (+)/(-)-LIM adsorption using powder GCN and GCN-CSA. **a**, (+)-LIM adsorption using GCN. **b**, (-)-LIM adsorption using GCN. **c**, (+)-LIM adsorption using GCN-CSA. **d**, (-)-LIM adsorption using GCN-CSA.

We agree with the statement “high enough loading of CSA in GCN would be required for expressing enantioselective permeation”. As described in the Experimental Section, the initial mass ratio of GCN to CSA is 1:5, which is the optimized ratio, actually. Because we found that further increasing the starting amount of CSA does not obviously increase the loading amount of CSA in GCN-CSA, as indicated by elemental analyses shown in Table R4. Also, the efficiency of enantioselective permeation shows no improvement when using higher initial mass ratio of GCN to CSA (1:10).

Table R4 Elemental analyses of GCN and GCN-CSA.

Sample	C (wt%)	N (wt%)	S (wt%)	C/N (molar ratio of GCN)
GCN	34.95	60.74	/	0.67
GCN-CSA (1:1)	33.29	45.35	1.998	0.66
GCN-CSA (1:5)	33.60	41.16	2.852	0.65
GCN-CSA (1:10)	33.49	40.96	2.885	0.65

Regarding the selective permeation (chiral separation) mechanism:

A permeation phenomenon is associated with a kinetic adsorption-diffusion-desorption process, involving the complicated interaction among solvent, solute and permeation material. Therefore, it is difficult to find out a universal explanation to address the chiral separation mechanism. A case-by-case analysis needs to be conducted according to the physical and chemical properties of solvent, solute and permeation material. In our work, we ascribe the selective permeation (chiral separation) of (+)-LIM over (-)-LIM to their steric effect and thus different interaction with chiral component of CSA in GCN-CSA membrane, which in turn causes the different adsorption behavior as described above and final selective permeation.

The relevant discussion has been supplemented into the main text (pages 9-11), Supplementary Section 8.3 (page 45) and Supplementary Section 8.4 (page 47).

5. Typos noticed;

Line 36; staking  stacking

Fig. 3; c  b, b  c

Fig. S19; e f, f e

Response: The typos have been corrected in the main text and Supplementary Information.

REVIEWERS' COMMENTS:

Reviewer #1 (Remarks to the Author):

The manuscript can be accepted for publication in its current form.

Reviewer #2 (Remarks to the Author):

The authors have properly addressed the issues, it because acceptable for publication.

Reviewer #3 (Remarks to the Author):

In this revised version, the calculation procedure of solvent permeability is well described with revised calculation results. Solute permeation rates were also well examined and solute concentration data were added on Supplementary. The response from the author about the molecular sieving performance of the amorphous GCN-CSA seems to be not so clear, but I understand that the author can share the concern with readers. I'd like to expect the authors future (next) work for revealing this issue. Newly measured and added adsorption data of LIM and GCN element analysis can be highly regarded. The authors' explanation and discussions of enantioselective permeation seems to be reasonable.

The authors have adequately addressed my questions, thus I highly appreciate the authors' work for improving this paper. As the result, the paper seems to be sufficiently of high quality. I would recommend it for publication in the journal as is.

Bo Liu, Ph. D., Professor
Department of Chemistry
University of Science & Technology of China (USTC)
96 Jinzhai Road, Hefei, Anhui 230026, P.R. China
Tel/ Fax: 86-551-63601123
Email: liuchem@ustc.edu.cn

Point-by-point responses to the referees' comments

(Referees' comments and the responses are displayed in black and blue, respectively)

Reviewer #1 (Remarks to the Author):

The manuscript can be accepted for publication in its current form.

Response: Thanks a lot for your time and effort on our manuscript.

Reviewer #2 (Remarks to the Author):

The authors have properly addressed the issues, it because acceptable for publication.

Response: Thanks a lot for your time and great effort on our manuscript.

Reviewer #3 (Remarks to the Author):

In this revised version, the calculation procedure of solvent permeability is well described with revised calculation results. Solute permeation rates were also well examined and solute concentration data were added on Supplementary. The response from the author about the molecular sieving performance of the amorphous GCN-CSA seems to be not so clear, but I understand that the author can share the concern with readers. I'd like to expect the authors future (next) work for revealing this issue. Newly measured and added adsorption data of LIM and GCN element analysis can be highly regarded. The authors' explanation and discussions of enantioselective permeation seems to be reasonable.

The authors have adequately addressed my questions, thus I highly appreciate the authors' work for improving this paper. As the result, the paper seems to be sufficiently of high quality. I would recommend it for publication in the journal as is.

Response: Thanks a lot for your time and effort on our revised manuscript. We highly appreciate the valuable comments that help us to improve the quality of this work.